# Immunoglobulin Superfamily Containing Leucine-Rich Repeat (Islr) Participates in IL-6-Mediated Crosstalk between Muscle and Brown Adipose Tissue to Regulate Energy Homeostasis

**DOI:** 10.3390/ijms231710008

**Published:** 2022-09-02

**Authors:** Chang Liu, Jin Liu, Tongtong Wang, Yang Su, Lei Li, Miaomiao Lan, Yingying Yu, Fan Liu, Lei Xiong, Kun Wang, Meijing Chen, Na Li, Qing Xu, Yue Hu, Yuxin Jia, Qingyong Meng

**Affiliations:** A State Key Laboratories for Agrobiotechnology, College of Biological Sciences, China Agricultural University, Yuanmingyuan West Road No. 2, Haidian District, Beijing 100193, China

**Keywords:** BAT-muscle cross talk, energy expenditure, Islr, muscle-derived IL-6, mitochondrial function, Ndufs2, thermogenesis

## Abstract

Brown adipose tissue (BAT) is functionally linked to skeletal muscle because both tissues originate from a common progenitor cell, but the precise mechanism controlling muscle-to-brown-fat communication is insufficiently understood. This report demonstrates that the immunoglobulin superfamily containing leucine-rich repeat (Islr), a marker of mesenchymal stromal/stem cells, is critical for the control of BAT mitochondrial function and whole-body energy homeostasis. The mice loss of Islr in BAT after cardiotoxin injury resulted in improved mitochondrial function, increased energy expenditure, and enhanced thermogenesis. Importantly, it was found that interleukin-6 (IL-6), as a myokine, participates in this process. Mechanistically, Islr interacts with NADH: Ubiquinone Oxidoreductase Core Subunit S2 (Ndufs2) to regulate IL-6 signaling; consequently, Islr functions as a brake that prevents IL-6 from promoting BAT activity. Together, these findings reveal a previously unrecognized mechanism for muscle-BAT cross talk driven by Islr, Ndufs2, and IL-6 to regulate energy homeostasis, which may be used as a potential therapeutic target in obesity.

## 1. Introduction

The global prevalence of obesity and associated metabolic diseases has risen to a level that now represents a major threat to human health [1,2]. As a key site for nonshivering thermogenesis, BAT dissipates excess fuel energy as heat, and the activities of BAT may be a target to prevent obesity and its related metabolic disorders [3,4]. Given that BAT and skeletal muscle originate from a common progenitor cell, they have closely related functions [5,6,7]. These precursors transiently express *Pax7* (Paired Box 7) and *Myf5* (Myogenic Factor 5). Then a part of the cells proliferate into Pax7^+^/MyoD^+^(Myogenic Differentiation^+^) myoblast cells [8] and differentiate into the MyoG^+^ (myogenin^+^) cells that fuse to form multinucleated myotubes [9]. The other part of the cells give rise to brown adipocytes. So the fate of daughter cells is decided in muscle direction and express both *Pax7* and *Myf5*, but only *Myf5* is expressed in the brown adipocyte direction. Satellite cells (SCs) play an important role during skeletal muscle regeneration, maintenance, and growth [10,11]. Cardiotoxin (CTX) had been suggested to cause cell fragmentation in skeletal muscle cells by forming pores in plasma membranes, and CTX-injury is a standard model for muscle regeneration and SCs activation in vitro [12,13]. As two of the most important metabolic organs of the human body, BAT and skeletal muscle can promote metabolic homeostasis by regulating glucose uptake and multiple interorgan communication [14,15].

Despite these similarities, few studies have investigated the communication between BAT and skeletal muscle in humans. A study with rodents showed that BAT cross talks with muscle and that BAT-specific knockout of Interferon Regulatory Factor 4 (IRF4) results in enhanced expression of Myostatin and other myogenic genes and thereby reduces the exercise capacity [16]. Another recent study found a 12,13-dihydroxy-9Z-octadecenoic acid (12,13-diHOME) signal from BAT to muscle was induced by exercise. This signal promotes fatty acid uptake in skeletal muscle, which suggests its role as a crucial lipid mediator in metabolic changes [17]. Beige adipocytes have a different origin from BAT; however, like BAT, beige adipocytes are highly adapted to expend chemical energy in the form of heat though the action of Uncoupling Protein 1 (Ucp1) [18]. Skeletal muscle also has endocrine functions and secretes myokines during exercise that can affect the browning of adipose tissues [19,20]. Irisin, a myokine, has the ability to promote the browning of white adipose tissue (WAT) and stimulates the expression of Ucp1 and other brown-specific genes [21]. Several studies have demonstrated that meteorin-like, a muscle-derived myokine, is induced after exercise, and this myokine stimulates beige fat thermogenesis-related gene expression and energy expenditure (EE) and improves glucose tolerance [22]. Thus, understanding the molecular mechanism though which myokines drive BAT thermogenic activity could be beneficial to the discovery of potential targets for the treatment of obesity.

Islr is an important marker of mesenchymal stromal/stem cells. Our group previously found that Islr is critical for the development and regeneration of skeletal muscle [23]. Subsequent studies have shown that Islr regulates obesity-induced metabolic disorders, mainly though the insulin signaling pathway. GO analysis showed Islr high expression in adipose tissue. These results raise the possibility that Islr participates in cross talk between skeletal muscle and adipose tissue.

In this study, using Islr gene whole-body knockout and conditional knockout mouse models, we examined the role of Islr-mediated dialog in muscle to BAT. Our results revealed an essential role of Islr in controlling the BAT response to muscle-derived IL-6 as well as metabolic homeostasis. The ablation of Islr in BAT resulted in a smaller lipid droplet size in the tissue, which led to enhanced EE, an improved body temperature and an increased number of BAT mitochondria after CTX treatment. Mechanistically, we demonstrated that Islr directly interacts with Ndufs2. Further experiments proved that Islr deletion leads to the accumulation of Ndufs2 and promotes thermogenesis in BAT under IL-6 stimulation. Critically, we also found that loss of Islr in BAT improved energy homeostasis in middle-aged mice with increases in IL-6. Thus, we uncovered Islr-dependent thermogenesis in BAT that communicates with skeletal muscle.

## 2. Results

### 2.1. Mice Lacking Islr Have Increased Whole-Body Energy Expenditure upon Muscle-BAT Dialogue

Previous studies have shown the important function of Islr in skeletal muscle and adipose tissue [23]. Whether muscle affects BAT function via Islr is unknown, and to investigate the communication between muscle and BAT controlled by Islr, we generated Islr-KO mice using the CRISPR-Cas9 system (Appendix A). Islr-knockout mice were born at the expected Mendelian ratios and appeared normal. The expression of Islr mRNA was markedly reduced in multiple adipose tissue types and muscle of the Islr-KO mice compared with those in their wild-type (WT) littermates (Appendix A). Unexpectedly, the morphology of BAT in the KO mice appeared deeply brown compared with that in WT mice after the tibial anterior (TA) muscles were injured via an injection of CTX, but no difference was observed under normal conditions (Figure 1a). We then examined the lipid accumulation in adipose tissue. An H&E analysis showed that Islr expression reduced the lipid droplet size of brown adipocytes (BAs), an indicator of active thermogenesis, in the BAT after CTX, and no obvious differences were found between the WT and KO uninjured mice. Quantification of lipid area also showed smaller BAs size in KO mice after CTX (Figure 1b). Likewise, the inguinal and gonad WAT adipocytes did not change (Appendix A). In addition, a thermal imaging scanner showed that the KO mice had markedly elevated core body temperatures after CTX injury (Figure 1c), which indicated an increase in thermogenesis, but an elevation was not detected in uninjured mice (Appendix A). These observations indicate that the KO mice have an altered energy homeostasis after CTX injury. To test the role of Islr in thermogenesis in vivo, we performed indirect calorimetric analyses under uninjured and CTX-injured conditions using metabolic cages. Under uninjured conditions, we found that the KO mice had oxygen consumption (VO_2_) (Appendix A) and EE (Figure 1d,e) levels similar to those of the WT mice. Consistent with the finding that the KO mice had an improvement in their core body temperature after CTX injury, significant increases in VO_2_ (Appendix A) and EE (Figure 1f,g) were observed in the KO mice during the CTX process. Importantly, no difference in body weight and locomotor activity were found in CTX-injured mice (Appendix A). These data indicate that the observed reduction in the brown adipocyte lipid droplet size resulted from an improvement in systemic EE caused by Islr deficiency after CTX injury.

### 2.2. Mice Lacking Islr in BAT Have Improved Energy Expenditure

We found that Islr-KO mice had improved energy expenditure, which was unexpected because the CTX injury was specifically to muscle, but we observed this phenotype in BAT. Therefore, we needed to determine if these effects happened in muscle and BAT. To this end, we generated skeletal muscle- and BAT-deleted Islr-mKO mice by breeding Islr-floxed mice (Islr^fl/fl^) with *Myf5*-Cre mice (Appendix A). The expression of Islr mRNA in BAT was markedly lower in mKO mice than in their mCtrl littermates (Appendix A), and the expression of Islr in muscle was significantly lower as previously reported [23]. As expected, the mKO mice had deeply brown BAT and smaller brown adipocyte lipid droplet sizes after CTX injury. Quantification of lipid area also showed smaller BAs size in mKO mice after CTX (Figure 2a). The rectal temperature in mKO mice was significantly higher than in mCtrl mice (Figure 2b). In addition, no difference in VO_2_ (Appendix A) or EE (Figure 2c,d) was observed under normal conditions. Notably, there were effective increases in VO_2_ (Appendix A) or EE (Figure 2e,f) after CTX treatment in KO mice.

With *Myf5*-Cre operating in both BAT and muscle, it is important to determine why this happens. Therefore, we then crossed skeletal muscle-specific Islr-deleted pKO mice by generating Islr-floxed mice (Islr^fl/fl^) with *Pax7*-Cre^ER^ mice (Appendix A). The mice were induced with tamoxifen at 8-weeks old and then subjected to CTX damage. To test whether the pKO mice had a phenotype consistent with that of the Islr-KO mice and mKO mice, we histologically analysed the BAT of the mice and found no difference in lipid droplet size (Figure 2g). In addition, no difference was observed in VO_2_ (Appendix A) or EE (Figure 2h) under CTX injury in pKO mice. These results indicate that a lack of Islr in BAT enhances EE during CTX injury.

### 2.3. Loss of Islr Enhances Mitochondrial Function in BAT

The increase in EE without an effect on food intake and locomotor activity leads to the hypothesis that a lack of Islr enhances thermogenesis by brown fat. A microscopic analysis found that the lipid droplets in BAT were smaller in the KO and mKO mice than in their littermate controls, whereas a higher mitochondrial density was observed in the KO mice (Figure 3a,b; Appendix A). To ascertain whether increased EE was associated with a mitochondrial activity in BAT, we extracted mitochondria from the BAT of the KO and WT control mice and measured the O_2_ consumption rate (OCR). The basal respiration and maximal respiration rates in mitochondria from CTX-injured KO mice were markedly higher than those in mitochondria from CTX-injured WT mice (Figure 3c). Consistent with this finding, we found greatly enhanced expression of thermogenesis proteins, UCP1, PGC-1α (PPARG Coactivator 1 Alpha), and mt-CO2 (Mitochondrially Encoded Cytochrome C Oxidase II), in CTX-injured KO mice (Figure 3d). In addition, Ucp1 and Pgc-1α were strongly expressed in BAT and mitochondria from mKO mice after CTX injury, which was consistent with mitochondrial activation (Appendix A). Even more importantly, we noted increased electron transport protein expression in BAT of KO mice after CTX injury (Figure 3e,f). A qPCR analysis of BAT of the KO-CTX mice revealed increased expression of a few key mitochondrial genes (Figure 3g,h,i). Overall, these results indicate that a lack of Islr after CTX injury enhances BAT mitochondrial function by activating mitochondrial activity.

### 2.4. Secreted Factors Promote Thermogenic Program in Islr-Deficient Brown Adipocytes

We then assessed whether skeletal muscle satellite cells (SSCs) from CTX-injured mice could directly affect thermogenic gene expression in BAs from KO mice by secreting myokines, adding SSCs growth and differentiation medium to mature BAs (Figure 4a). The Islr mRNA levels were lower in the BAs of Islr-KO mice, and we noted elevated expression of thermogenic genes (e.g., Ucp1, Pgc-1α, and Tfam (Transcription Factor A, Mitochondrial)) in isolated BAs from the KO mice after the addition of SSCs medium (KO + medium) (Figure 4b). As predicted, the expression of mitochondrial key genes (Figure 4c) and electron transport chain (ETC) genes (Figure 4d) in KO+medium exhibited patterns similar to those in BAT of KO-CTX mice. In addition, the mitochondrial protein levels were increased in KO + medium (Figure 4e,f). These results indicate that SSCs from CTX-injured mice can secrete one or more myokines to promote thermogenesis in BAs from the KO mice.

### 2.5. IL-6 Was Identified as an Essential Myokine That Promotes BAT Thermogenesis

In response to the findings given above, we performed proteomics analyses of TA muscle derived from the normal and CTX groups. Subsequently, we chose candidates that were highly expressed in the CTX group relative to the normal group and contained a secretory signal peptide. A bioinformatics analysis identified 16 highly secreted candidates from the CTX group (Figure 5a). To screen the metabolic roles of these candidates, we then used recombinant protein to treat the differentiated SVF of BAT from the WT and KO mice (given as WT and KO BATSVF below). Notably, recombinant IL-6 promoted Ucp1 expression in the KO BATSVF compared to the other groups (Figure 5b). IL-6 was further investigated because a previous study also found an increase in IL-6 following injury, particularly on the fifth day [24].

If IL-6 has an important role in thermogenesis, IL-6 supplementation mimics some of the metabolic effects of CTX injury. We confirmed that the injection of IL-6 in the BAT from the KO mice significantly reduced the lipid droplet size of BAs in the tissues. Quantification of the lipid area showed the same results (Figure 5c). Consistent with the decreased lipid droplet size, we observed increased levels of thermogenic protein, including UCP1, NDUFS2, TOM20 (Translocase of Outer Mitochondrial Membrane 20), and TFAM, in the BAT of IL-6-treated mice (Figure 5d), which indicated that IL-6 promotes BAT thermogenesis. By inference from these results, the treatment of differentiated BATSVF isolated from the KO mice with recombinant IL-6 significantly increased the expression of mitochondrial key genes (Figure 5e) and ETC genes (Figure 5f) in comparison to the other groups. In addition, KO BATSVF treated with IL-6 exhibited greater OCR as evidenced by elevations in the basal respiration and maximal respiration rates (Figure 5g). To determine if the improvement in thermogenesis induced by IL-6 in KO BATSVF is due to activation of the IL-6 signaling pathway, we measured thermogenesis genes, Ucp1 and Pgc-1α, after treatment with an anti-IL-6 receptor antibody (tocilizumab) and IL-6. We observed that the increase in Ucp1 and Pgc-1α expression triggered by IL-6 had virtually disappeared in the KO BATSVF treated with tocilizumab (Figure 5h). Despite its correlative nature, these latter data indicate that IL-6 may act as a myokine that participates in CTX injury and increases systemic EE.

### 2.6. Ndufs2 Interacts with Islr and Is Required for the IL-6 Mediates Thermogenesis

The above data led us to explore the molecular mechanism underlying the enhanced thermogenic function of BAT in Islr-KO mice. To test the model in detail, we asked whether mitochondrial genes are needed for Islr to control energy homeostasis. Consequently, we combined proteomics data with our yeast two-hybrid database to predict Islr interacted proteins (Appendix A). We chose candidate mitochondrial genes that participate in the thermogenic pathway. Ndufs2 justified further investigation because of its negative relationship with Islr (Appendix A). The strong upregulation of Ndufs2 expression in the absence of Islr was confirmed at the protein level in BAT (Appendix A). Moreover, the downregulation of Ndufs2 expression in Islr-overexpression mice was confirmed at the protein level in BAT (Appendix A). In addition, we extracted cytosol and mitochondria from the BAT of the mKO and mCtrl control mice, and Ndufs2 markedly accumulated in mitochondria in the CTX-injured mKO mice (Appendix A). We cotransformed the pGBKT7-Islr and pGADT7-Ndufs2 plasmids and grew them on QDO/X/A plates. Blue clones were observed on QDO/X/A plates, which indicated that Ndufs2 interacted with Islr (Figure 6a). To further determine whether Islr interacted with Ndufs2, we performed IP analysis and found that Islr-GFP interacted with Ndufs2-GFP, and Ndufs2-GFP interacted with Islr-GFP (Figure 6b). To further elucidate the mechanism of Ndufs2 upregulation, the Ndufs2 protein levels in WT and Islr-KO BATSVF cells were examined after cycloheximide (CHX) or CHX + MG132 was used to inhibit de novo protein synthesis or proteasomal degradation, respectively. When protein synthesis was inhibited by CHX, the Ndufs2 protein level decreased rapidly in WT BATSVF, but the Ndufs2 protein level remained high in KO BATSVF (Figure 6c,d), which indicated that the protein stability of Ndufs2 was increased in Islr-KO cells. However, when proteasomal degradation was inhibited by MG132 for 6 h to determine if there was a difference in protein synthesis, the Ndufs2 level remained steady in WT and KO BATSVF (Figure 6e,f). These results indicate that Islr-KO increases the protein expression of Ndufs2 by inhibiting its proteasomal degradation.

We then considered if Ndufs2 mediates the observed biological function of Islr. We observed that the KO mice injected with IL-6 had increased Ndufs2 protein levels (Figure 5d). Then, using a mature BA transfection technique [25], we knocked down Ndufs2 in both differentiated WT and Islr-KO BATSVF cells with siRNA (Appendix A). The inhibition of Ndufs2 markedly reduced the Ucp1 protein and mRNA levels induced by IL-6 treatment in KO differentiated SVF (Figure 6g,h), and an increase in OCR induced by IL-6 treatment in KO differentiated SVF was also effected by Ndufs2 knockdown (Figure 6i). These data indicate that the loss of Islr promotes thermogenesis activity in BAT by interacting with the mitochondrial gene Ndufs2 and that Ndufs2 is needed for IL-6-mediated thermogenesis.

### 2.7. Deletion of Islr Exerts Beneficial Metabolic Effects in Middle-Aged Mice

In our study, we found that IL-6 was a key myokine affecting BAT after muscle CTX injury. Even more importantly, loss of Islr in BAT had a greater effect on BAT activity. To ascertain if increased IL-6 was associated with an activity in BAT of Islr-deficient mice, we used a middle-aged mouse model. Previous studies demonstrated that aging is related with an elevation in the levels of IL-6 [26,27]. We next examined the metabolic phenotypes of elderly mCtrl and mKO mice fed with a chow diet. We found that mKO mice had significantly lower body weights compared to mCtrl mice (Figure 7a), and the KO mice also had reduced body weights compared to WT mice (Appendix A). Histologically, the morphology of BAT in the mKO and KO mice was a deeper brown than in their littermate controls, and we also found that BAT of mKO and KO mice had a reduced adipocyte lipid droplet size, an indicator of active thermogenesis (Figure 7b and Appendix A). Indeed, the serum IL-6 levels of middle-aged mice were higher than young mice (Figure 7c). In addition, the mKO and KO mice had a higher rectal temperature than the mCtrl and WT mice (Figure 7d and Appendix A). Subsequently, we found that the mKO mice had significant increases in VO_2_ (Figure 7e,f) and EE (Figure 7g,h). However, there was no difference in locomotor activity during the light and dark (Appendix A). Additionally, we subjected the mCtrl and mKO mice to a glucose tolerance test (GTT) and an insulin tolerance test (ITT). The mKO mice displayed a slightly higher lower glucose tolerance relative to mCtrl mice at 15 min; however, we did not detect any differences at other time points (Figure 7i). Moreover, mKO mice displayed increased insulin sensitivity compared with mCtrl mice (Figure 7j). Together, these results indicate that Islr-KO mice have improved middle-aged metabolic homeostasis.

## 3. Discussion

Emerging evidence shows that BAT is not only important in thermogenesis, but it also regulates glucose and lipid homeostasis [28,29]. However, communication between BAT and peripheral metabolic organs, such as the brain, the liver, and skeletal muscle, has been reported [16,30,31,32]. In this study, we demonstrated that Islr, an important marker of mesenchymal stromal/stem cells, controls the communication between BAT and skeletal muscle via IL-6. We found that loss of Islr in BAT increased BAT thermogenesis and EE via IL-6 after CTX injury. We further obtained both in vitro and in vivo evidence that IL-6-induced systemic energy changes under Islr-deficient conditions were accompanied by enhanced mitochondrial function, which in turn stimulated Ndufs2 accumulation in BAT. We also obtained evidence supporting the finding that Ndufs2 interacts with Islr and that Ndufs2 is needed for the IL-6 to mediate the effect of Islr loss on thermogenesis. Previous studies have revealed that Islr is necessary for the regulation of heart repair and fibrosis, the development and regeneration of skeletal muscle, and the obesity-induced insulin resistance [23,33]. This work adds Islr as a novel regulator controlling muscle–BAT dialogue.

We showed that the effects on BAT activity were observed only in the muscle injury model and in aged mice, which raised the question if indirect effects, such as the inflammatory reaction or housed ambient temperature, could account for the observed phenotype. First of all, we used three mouse models in our research, and these models, including control mice and knockout mice, were all subjected to CTX injury and housed in the same environment. However, comparing with the control mice, all the mice were exposed to an inflammatory environment and had the same background. We thought the inflammatory reaction had an effect on this phenotype, for example IL-6 in this work, but the inflammatory environment could be the same background. Consequently, both knockout Islr in BAT and injury in muscle, present in this process, are key factors. We conclude the deletion of Islr in BAT was the most important.

IL-6 exerts its biological activities as a multifunctional cytokine. Toshio et al., claimed that when the body is disordered by infections or tissue injuries, IL-6 is produced immediately and contributes to the host adapting such emergent stress via activation of an immune responses [34]. In this study, we demonstrated that KO mice increased EE after muscle injury via IL-6, which may improve the host immune system. IL-6 is also thought to be involved in exercise-induced WAT browning because the expression of marker genes for this process is inactivated in IL-6-deleted mice [35]. Through the injection of TA CTX and BAT localized injection of IL-6, we showed that the WT-CTX group had a smaller lipid droplet size than the WT normal group, and this finding was consistent with the results obtained with IL-6 injection. Importantly, we found the KO-CTX group enhanced this phenotype, and the injection of IL-6 into KO-BAT enhanced this phenotype. These results further support the proposition that IL-6 mediates browning and that a lack of Islr exerts beneficial effects on BAT activation.

In this study, although no significant changes in body weight were found in WT and KO mice after CTX injury, a slight decrease was found in the KO mice. This finding could be explained by the fact that the KO mice had increased EE. Indeed, the mKO mice had decreased in body weight with middle-aged and increased EE. Aging also increases the levels of proinflammatory cytokines, such as IL-6 [36]. It appears that Islr, as an important effector of the IL-6 signaling, antagonizes IL-6-induced WAT or BAT browning, and loss of Islr enhances IL-6-activated thermogenesis.

Ndufs2 is well known as an NADH ubiquinone oxidoreductase and is mainly expressed in the mitochondrial inner membrane; this complex is the largest complex in mitochondria and contributes to thermogenesis [37]. Mutations in Ndufs1/Ndufs2 result in mitochondrial complex I deficiency in various disorders of mitochondrial oxidative phosphorylation (OXPHOS) [38,39,40], supporting the proposition that Ndufs2 upregulation in BAT of KO-CTX mice increases the OXPHOS protein levels. Mitochondria are double-membraned organelles. Mitochondrial outer-membrane proteins are degraded by the ubiquitin-proteasome system, and the proteostasis of intramitochondrial compartments is maintained by AAA proteases [41,42,43]. Islr promotes the degradation of Ndufs2 though the proteasome pathway, which indicates that Islr might be a substrate-binding site needed for the degradation of Ndufs2. A recent study showed that oncostatin M receptor (OSMR) interacts with Ndufs1/Ndufs2 and promotes mitochondrial respiration [44]. OSMR is a member of the IL-6 receptor family that regulates homeostasis and cell growth and differentiation [45,46]. The major findings of our study indicate that Islr interacts with Ndufs2. Islr may competitively bind Ndufs2 via OSMR. In the absence of Islr, Ndufs2 interacts with OSMR and gives a greater response to muscle-derived IL-6, which increases the OXPHOS levels and promotes mitochondrial respiration. Future studies on Islr, Ndufs2, and OSMR are merited.

Briefly, some myokines, such as IL-6, can regulate the BAT metabolism. This cross talk or interplay between skeletal muscle and BAT maintains metabolic homeostasis. Islr may competitively bind Ndufs2 and block IL-6 signaling to subtly regulate crucial energy consumption in BAT. These results indicate a novel metabolic mechanism of homeostasis regulation and related aging health control to provide recommendations for the future treatment of metabolic-related diseases.

## 4. Materials and Methods

### 4.1. Animals

All animal experiments used male mice, and they were housed in a specific pathogen-free animal facility in a controlled environment (12 h light/dark cycle; humidity 50–60%; ambient temperature 22 ± 2 °C). The type of food supplied to the mice was rat and mice reproduction feed (SPF-F01-002, Sipeifu, Beijing, China). To generate conditional Islr knockout mice (mKO or pKO), floxed Islr (Islrfl/fl) mice were crossed with *Myf5*-Cre and *Pax7*-Cre^ER^ mice on a C57BL/6 background. Details on the *Myf5*-Cre and *Pax7*-Cre^ER^ transgenic mice and Islrfl/fl mice and their genotypes were determined by PCR as described previously [23]. The global Islr knockout mice (KO) mice generated by Cyagen on a C57BL/6 background, and their genotype was determined by PCR, using the following primer pairs: Islr-KO forward: 5′-ggttggctataaagaggtcatcag-3′; Islr-KO reverse: 5′-agcaactagcactagagatgatgag-3′; WT forward: 5′-tttgttgggctgttctcctg-3′; and WT reverse: 5′-catcttggctgggctggta-3′. Animals experiments used male mice having a similar body weight. CTX-induced muscle regeneration was established as described previously [23]. Briefly, 10-week-old mice were anaesthetized with isoflurane; mouse legs were shaved and cleaned with alcohol. A total of 100 μL of 10 μM CTX (Sigma-Aldrich, C9759) solution was injected into the TA to cause muscle injury for the muscle regeneration experiments to explore muscle-BAT dialogue. GTT and ITT were performed at 10 months of middle-aged mice. Housing, husbandry, and all experimental protocols used in this study were performed according to the Regulations of Beijing Laboratory Animal Management and were approved by the China Agricultural University Laboratory Animal Welfare and Animal Experimental Ethical Inspection Form (approval number: AW12906102-3-1).

### 4.2. H&E Staining

The adipose tissues were fixed in 4% (*v*/*v*) PFA for more than 48 h at room temperature and embedded in paraffin. In brief, the tissues were dehydrated, incubated in xylene, paraffin-embedded (DAKEWE), and sectioned at 3.5 μm using a microtome (Leica). The tissue sections were then deparaffinized, hydrated, washed, stained with hematoxylin and eosin, washed, dehydrated, cleared, and mounted. To quantify adipocyte size in BAT, H&E staining quantified with Fiji software. Data from three to six mice from each group were averaged.

### 4.3. Mitochondrial Isolation

Mitochondria were isolated from BAT as previously described [47]. Briefly, BAT was minced and homogenized with a glass Teflon pestle, and mitochondria were isolated by differential centrifugation. The homogenates were centrifuged at 1000× *g* and 4 °C for 5 min, and the resulting supernatant was centrifuged at 3500× *g* and 4 °C for 10 min. The supernatant was removed carefully, and the pellet containing the mitochondria was used for further research. All procedures were conducted at 0–4 °C.

### 4.4. Western Blot Analysis

BAT or BATSVF lysates were extracted with RIPA buffer (CST) with 1% PMSF. Total protein lysates (30 μg) were loaded into a 10% SDS-PAGE gel and transferred to a PVDF membranes (0.45 μm, Millipore, America). The membrane was blocked in 5% skim milk for 1 h at room temperature and incubated with primary antibodies overnight at 4 °C and then with the corresponding HRP-labelled secondary antibodies (1:10,000) for 1 h at room temperature. The levels of β-tubulin and GAPDH served as the loading controls.

For IP analysis, transfected HEK293T cells were extracted with IP buffer (Beyotime Biotechnology, Shanghai, China), and the total protein was immunoprecipitated with Islr or Ndufs2 antibody and protein A/G beads (Thermo Fisher, Waltham, MA, USA). Cultured cells were treated with 20 μg/mL cycloheximide (CHX) (SC0353, Beyotime Biotechnology, Shanghai, China) or 20 μM MG132(M7449, Sigma-Aldrich, Darmstadt, Germany) for the indicated times and then lysed in RIPA buffer containing phosphatase inhibitors on ice for 10 min. The protein antibodies are described in Appendix A.

### 4.5. Quantitative Real-Time PCR Analysis

Total RNA from mouse BAT or adipocytes was extracted using TRIzol (Invitrogen, Waltham, America). RNA (1.5 μg) was used for reverse transcription to obtain cDNA (5X All-In-One RT MasterMix with AccuRT, Abm, Vancouver, BC, Canada). The expression levels of genes were analysed using a quantitative real-time PCR system (ABI StepOnePlus, Waltham, MA, USA) with 2X RealStar Green Power Mixture (GenStar, Beijing, China). The results were normalized to the expression of the housekeeping gene 36b4. The qRT-PCR primers are listed in Appendix A.

### 4.6. Brown Preadipocyte Preparation, Culture, and Transfection

SVF cells were isolated from interscapular brown fat as previously described [48]. In brief, BAT was dissected from newborn mice, minced, and digested for 2 h at 37 °C with collagenase type II (2 mg/mL). Subsequently, the cells were filtered through a 100 μm cell strainer to remove undigested tissue and centrifuged for 5 min at 800× *g* to remove supernatant. The pellet containing SVF was resuspended in culture medium for further research. BA differentiation was induced by treating confluent SVF cells with differentiation medium (DMEM containing 10% FBS, 5 μg/mL insulin, 1 μM dexamethasone, 0.5 mM isobutylmethylxanthine, 1 nM T3, 125 μM indomethacin, and 1 μM rosiglitazone) for 2 days. The cells were then cultured with maintenance medium (DMEM containing 10% FBS, 5 μg/mL insulin, 1 nM T3, and 1 μM rosiglitazone) for 2 days and then in DMEM containing 10% FBS for 2 days.

Mature BAs were transfected for the indicated measurements [25]. The sequences of siNdufs2 primers are forward 5′-GCACCAGGACCUACCUCUUTT-3′ and reverse 5′-AAGAGGUAGGUCCUGGUGCTT-3′.

### 4.7. Identification of Secreted Proteins in Mouse Muscle after CTX Injury

The 10-week-old male C57BL/6 mice were divided into two groups: the CTX- injured group and a control group. Briefly, 30 mg TA samples from mice 5 days after CTX injury and control mice were collected, adding 500 μL RIPA buffer (CST) with 1% PMSF, 10% PhosSTOP^TM^ (Roche), fully ground and homogenized. Then, samples were bathed in ice for 10 min and centrifuged for 15 min at 4 °C and 13000× *g* to separate debris or a lipid layer. Each 400 μL sample was mixed with 1600 μL pre-chilled acetone. After incubation at 20 °C overnight, the mixture was centrifuged at 13000× *g* for 25 min at 4 °C to remove supernatant and stored at −80 °C for testing. Tissue extracts were analyzed using LC-MS/MS analysis (ThermoFisher Q-Exactive). All myokine candidates were identified from a proteomics study of uninjured and CTX-injured TA muscles. Candidates were selected based on the criteria that their expression was enriched in CTX-injured TA relative to uninjured TA with a *p* value of <0.05.

### 4.8. Treatment of Cell Cultures of Mice with IL-6

Cells were differentiated for 6 days and then treated with recombinant IL-6 (50 ng/μL) or 0.1% BSA for 6 h. After IL-6 treatment, the cells were washed with PBS and collected for further research. For delivery into BAT, 10-week-old male mice were anesthetized with isoflurane, and their interscapular hair was shaved to expose the BAT area. To distribute recombinant protein to as much of the depot as possible, each BAT depot received 500 ng of IL-6 or 0.9% NaCl at different points. Six hours after injection, the mice were euthanized by cervical dislocation for BAT collection.

Seahorse: the cellular OCR was measured using an XFe24 Analyzer (Agilent Technologies, Palo Alto, America). Primary adipocytes were differentiated for 2 days and plated at 10,000 cells/well in an XFe24 cell culture microplate (Agilent Technologies, Palo Alto, America) following trypsin digestion. The adherent cells were differentiated for 6 days and then treated with vehicle or IL-6 (50 ng/μL) for 6 h. Adipocytes were incubated in Seahorse XF Base Medium (1 mM sodium pyruvate, 2 mM L-glutamine, and 25 mM glucose) for 1 h at 37 °C without CO_2_ before analysis. A mitochondrial stress test was then performed by injecting oligomycin (5 μM), carbonyl cyanide 4-(trifluoromethoxy) phenylhydrazone (FCCP, 5 μM) and rotenone/antimycin A (AA, 1 μM). For mitochondrial seahorse analysis, mitochondria isolated from BAT (5 μg) were placed into an XFe24 cell culture microplate and centrifuge at 4 °C and 2000× *g* for 20 min. The OCR was adjusted by the protein amount or mitochondrial weight.

### 4.9. Metabolic Cages

The VO_2_, carbon dioxide production (VCO_2_) and EE were determined with an Oxylet System (Panlab, Barcelona, Spain) and the METABOLISM software suite. Data were analysed as previously described [49]. In brief, the mice were placed in separate cages and adapted to the system for 2 days before being official tested with a free module of food and water throughout the process. The VO_2_, VCO_2_, and EE were analysed using METABOLISM software (version 3.0, harvard apparatus, Cambridge, MA, USA). The core body temperatures of the mice were surveyed with a probe thermometer.

### 4.10. In Vivo Metabolic Assays

Prior to the tests, the mice were fasted overnight (for the GTT) or for 6 h (for the ITT). For the GTT, the mice were injected with 2 g/kg of D-glucose. For the ITT, the mice received an intraperitoneal injection of insulin at a dose of 0.8 U/kg. The blood glucose levels were determined at the indicated time points using blood glucose test strips (ACCU-CHEK). Mice blood was collected to assess plasma, letting it stand still for 30 min at room temperature, centrifuged at 1000× *g* and 4 °C for 15 min. Then an ELISA kit was used to measure plasma IL-6 levels.

### 4.11. Statistics

A minimum of 3 and up to 10 replicates were performed for all experiments. Values are displayed as the means ± SEM. Statistical analyses were performed using Prism 9 (GraphPad). An unpaired two-tailed Student’s *t*-test was used to determine the statistical significance of the differences. For all analyses, * *p* < 0.05, ** *p* < 0.01, and *** *p* < 0.001.

## Figures and Tables

**Figure 1 ijms-23-10008-f001:**
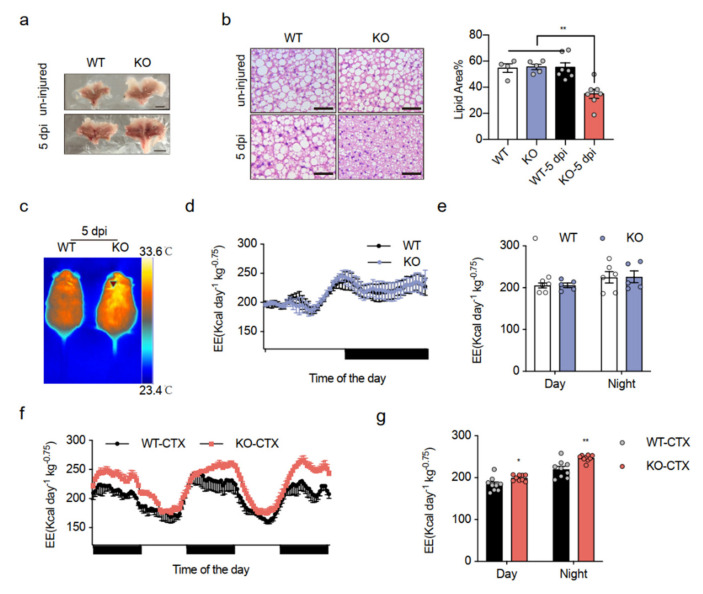
Mice lacking Islr have increased whole-body energy expenditure upon muscle-BAT dialogue: (**a**) representative BAT from WT and KO mice at the age of 10 weeks under uninjured and CTX-injured conditions; scale bar: 0.5 cm; (**b**) Left: histological images of BAT from WT and KO mice at the age of 10 weeks under uninjured and CTX-injured conditions (n = 6); Scale bar: 50 μm; Right: quantification of H&E staining under uninjured and injured conditions; (**c**) thermal images of WT and KO mice after CTX injury; (**d**,**e**) indirect calorimetric analysis of EE (**d**) and quantification of EE (**e**) of WT and KO mice under uninjured conditions (n = 5–6); (**f**,**g**) indirect calorimetric analysis of EE (**f**) and quantification of EE (**g**) of WT and KO mice after CTX injury (n = 9); WT: wild-type mice; KO: Islr-KO mice; 5 dpi: 5 days post injury CTX: cardiotoxin.; BAT: brown adipose tissue; EE: energy expenditure. Error bars represent SEMs, * *p* < 0.05, ** *p* < 0.01, as determined by two-tailed Student’s *t*-test.

**Figure 2 ijms-23-10008-f002:**
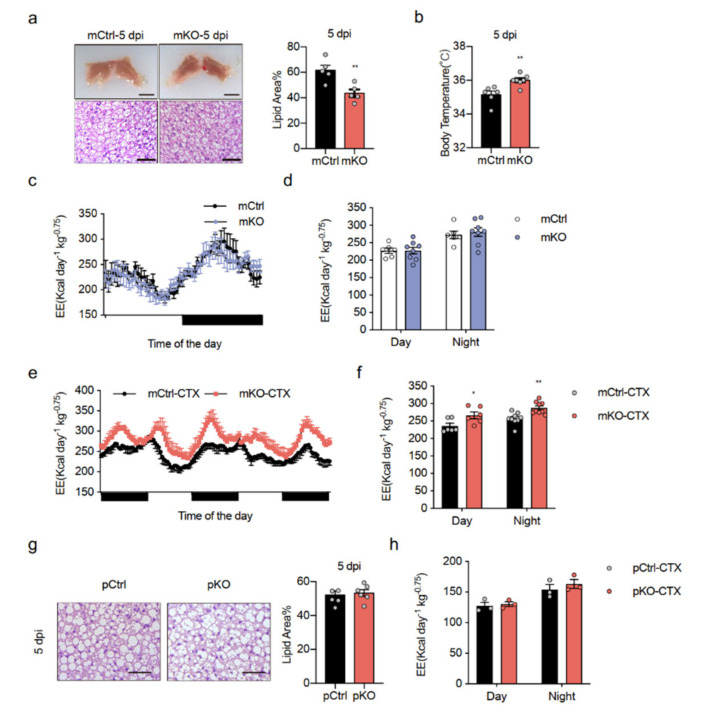
Mice lacking Islr in BAT exhibit increased energy expenditure: (**a**) Left: representative BAT tissues and H&E staining of BAT sections from mCtrl and mKO mice after CTX injury (n = 5); scale bar: 0.5 cm and 50 μm; Right: quantification of H&E staining under injured conditions; (**b**) quantification of the core body temperatures of mCtrl and mKO mice after CTX injury (n = 6–7); (**c**,**d**) indirect calorimetric analysis of EE (**c**) and quantification of EE (**d**) of mCtrl and mKO mice under normal conditions (n = 6–8); (**e**,**f**) indirect calorimetric analysis of EE (**e**) and quantification of EE (**f**) of mCtrl and mKO mice after CTX injury (n = 6–9); (**g**) histological images of BAT from pCtrl and pKO mice after CTX injury; scale bar: 50 μm; (left) quantification of H&E staining shown in the left panel (n = 6) (right), (**h**) quantification of EE of pCtrl and pKO mice after CTX injury (n = 3); mCtrl: Islr-floxed mice; mKO: BAT- and muscle-specific Islr-knockout mice, *Myf5*-Cre; Islr^fl/fl^; pCtrl: Islr-floxed mice; pKO: muscle-specific Islr-knockout mice, *Pax7*-Cre^ER^; Islr^fl/fl^. Error bars represent SEMs, * *p* < 0.05, ** *p* < 0.01, as determined by two-tailed Student’s *t*-test.

**Figure 3 ijms-23-10008-f003:**
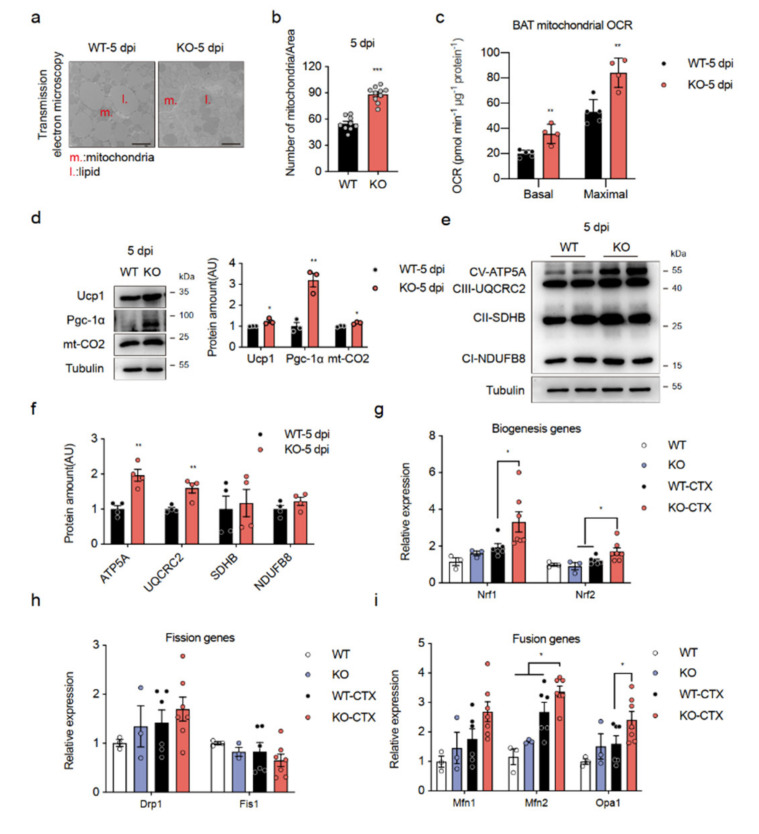
Loss of Islr enhances mitochondrial function in BAT: (**a**) TEM images of BAT from WT and KO mice after CTX injury; scale bar: 2.5 μm; (**b**) number of mitochondria per image of BAT of WT and KO mice (n = 9–10); (**c**) average basal and maximal BAT mitochondrial OCR of BAT from CTX-injured WT and KO mice (n = 4–5); (**d**) Western blots for UCP1, PGC-1α, and mt-CO2 in BAT of WT and KO mice after CTX injury; β-tubulin was used as a loading control; (**e**) Western blots showing mitochondrial oxidative phosphorylation (OXPHOS) proteins in BAT of WT and KO mice after CTX injury; β-tubulin was used as a loading control; (**f**) Quantification of the OXPHOS protein/β-Tubulin signal ratios normalized (set to 1) to those of the WT (n = 4); (**g**,**h**,**i**) expression of genes related to mitochondrial biogenesis (**g**), fission (**h**), and fusion (**i**) in BAT from WT and KO mice under uninjured and CTX-injured conditions (n = 3–7). Error bars represent SEMs, * *p* < 0.05, ** *p* < 0.01, and *** *p* < 0.001, as determined by two-tailed Student’s *t*-test. ATP5A: ATP Synthase F1 Subunit Alpha; UQCRC2: Ubiquinol-Cytochrome C Reductase Core Protein 2; SDHB: Succinate Dehydrogenase Complex Iron Sulfur Subunit B; NDUFB8: NADH:Ubiquinone Oxidoreductase Subunit B8; Nrf1: Nuclear Respiratory Factor 1; Nrf2: Nuclear Respiratory Factor 2; Drp1: Dynamin 1 Like; Fis1: Fission, Mitochondrial 1; Mfn1: Mitofusin 1; Mfn2: Mitofusin 2; Opa1: OPA1 Mitochondrial Dynamin Like GTPase.

**Figure 4 ijms-23-10008-f004:**
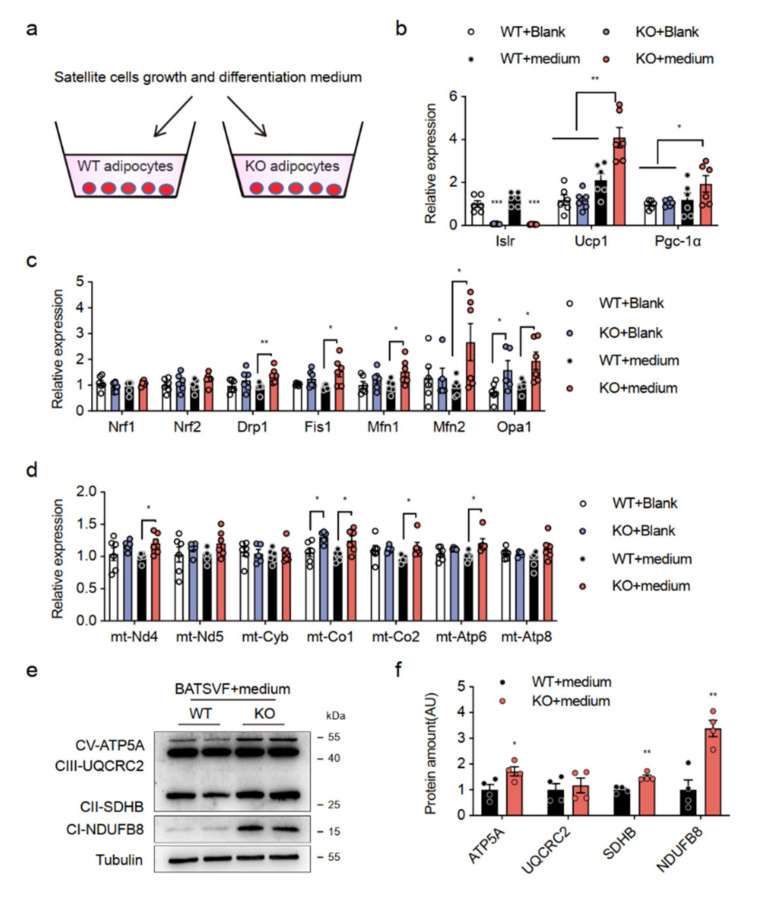
Secreted factors promote thermogenic program in Islr-deficient brown adipocytes; (**a**) schematic illustration of the cell culture process; (**b**) gene expression (thermogenesis genes) in WT and KO differentiation BATSVF treated with blank or skeletal satellite cell growth and differentiation medium (n = 6); (**c**,**d**) gene expression (mitochondrial genes and ETC genes) in WT and KO differentiation BATSVF treated with blank or skeletal satellite cell growth and differentiation medium (n = 5–6); (**e**) Western blot analysis of OXPHOS proteins in treated differentiation BATSVF; (**f**) Protein levels were quantified using Image J (n = 4). Error bars represent SEMs, * *p* < 0.05, ** *p* < 0.01, and *** *p* < 0.001, as determined by two-tailed Student’s *t*-test. mt-Nd4: Mitochondrially Encoded NADH:Ubiquinone Oxidoreductase Core Subunit 4; mt-Nd5: Mitochondrially Encoded NADH:Ubiquinone Oxidoreductase Core Subunit 5; mt-Cyb: Mitochondrially Encoded Cytochrome B; mt-Co1: Mitochondrially Encoded Cytochrome C Oxidase I; mt-Co2: Mitochondrially Encoded Cytochrome C Oxidase II; mt-Atp6: Mitochondrially Encoded ATP Synthase Membrane Subunit 6; mt-Atp8: Mitochondrially Encoded ATP Synthase Membrane Subunit 8.

**Figure 5 ijms-23-10008-f005:**
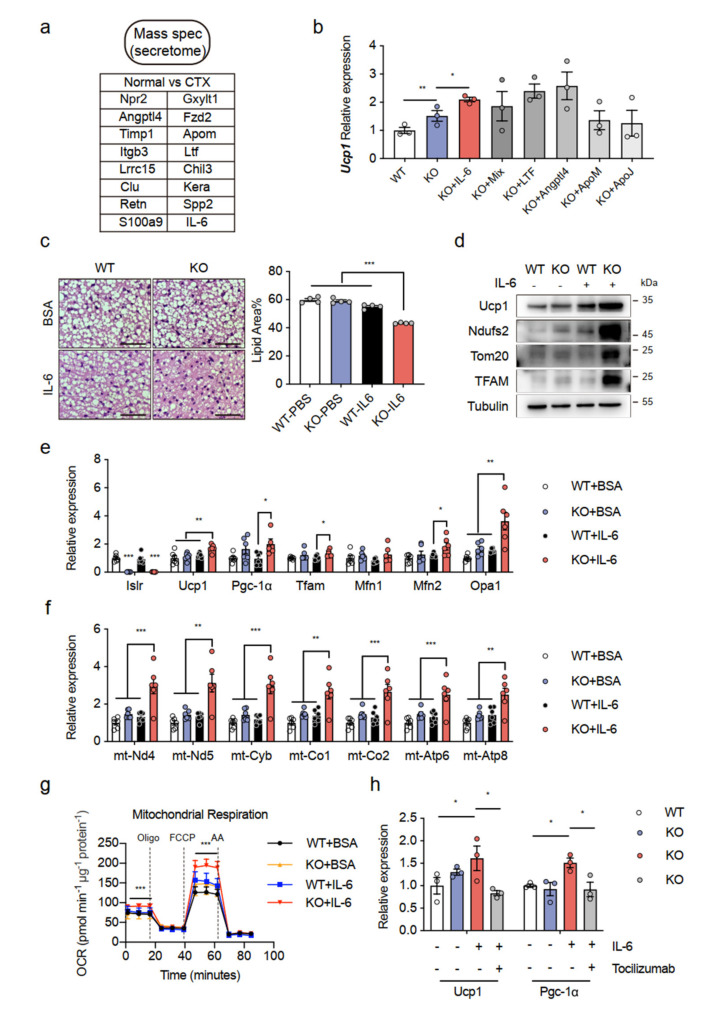
Induction of IL-6 as an essential myokine for promotion of BAT thermogenesis: (**a**) bioinformatic analysis defined 16 secreted candidates from CTX-injured TA compared with uninjured TA; (**b**) gene expression of Ucp1 in WT and KO BATSVF treated with candidate recombinant proteins; (**c**) histological images of BAT from WT and KO mice treated with BSA or IL-6 (n = 4); scale bar: 50 μm; (**d**) Western blot analysis of UCP1, NDUFS2, TOM20, and TFAM in BAT of WT and KO mice treated with BSA or IL-6; β-Tubulin was used as a loading control; (**e**,**f**) gene expression (mitochondrial genes and ETC genes) in WT and KO BATSVF treated with BSA or IL-6 (n = 6); (**g**) OCR in WT and KO BATSVF treated with BSA or IL-6; oligomycin (oligo), FCCP, and rotenone/antimycin (AA) were added at the time points indicated by dashed lines (n = 4); (**h**) gene expression in WT and KO BATSVF treated with BSA, IL-6, or a neutralizing antibody against the IL-6 receptor (tocilizumab) (n = 3). Error bars represent SEMs, * *p* < 0.05, ** *p* < 0.01, and *** *p* < 0.001, as determined by two-tailed Student’s *t*-test. Npr2: natriuretic peptide receptor 2; Gxylt1: glucoside xylosyltransferase 1; Angptl4: angiopoietin-related protein 4; Fzd2: frizzled-2; Timp1: metalloproteinase inhibitor 1; Apom: apolipoprotein M. Itgb3: integrin beta-3; Ltf: lactotransferrin; Spp2: secreted phosphoprotein 2; Lrrc15: leucine-rich repeat-containing protein 15; Chil3: chitinase-like protein 3; Clu: clustering; Kera: keratocan; Retn: resistin; S100a9: protein S100-A9.

**Figure 6 ijms-23-10008-f006:**
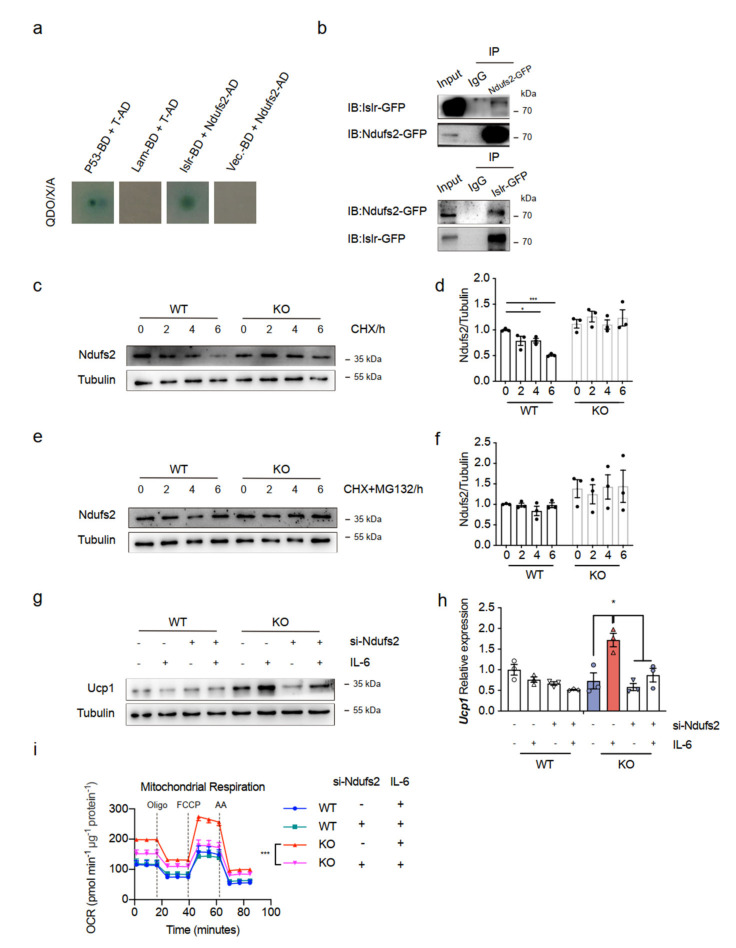
Ndufs2 directly interacts with Islr and is needed for IL-6-mediated thermogenesis: (**a**) Y2HGold cells were cotransformed with pGBKT7-Islr and pGADT7-Ndufs2 and plated on QDO/X/A plates; a blue color indicated a positive interaction. Cotransformation with pGBKT7-p53 and pGADT7-T was used as a positive control, cotransformation with pGBKT7-lam and pGADT7-T was used as a negative control, and cotransformation with pGADT7-Ndufs2 and pGADT7-T was used as a blank control; (**b**) reciprocal coimmunoprecipitation(co-IP) analysis of GFP-tagged Islr and GFP-tagged Ndufs2 in HEK293T cells; IB: immunoblotting; IP: immunoprecipitation; (**c**,**d**) Western blot analysis of Ndufs2 in WT and KO BATSVF treated with CHX for the indicated times (**c**); quantification of the Ndufs2 degradation rate by grayscale analysis (**d**); (**e**,**f**) Western blot analysis of Ndufs2 in WT and KO BATSVF treated with CHX+MG132 for the indicated times (**e**); quantification of the Ndufs2 degradation rate by grayscale analysis (**f**); (**g**,**h**) Western blot (**g**) and qPCR (**h**) analyses of Ucp1 in differentiated BATSVF (day 6) upon Ndufs2 siRNA or IL-6 treatment (n = 3); (**i**) OCR of differentiated BATSVF cells treated with Ndufs2 siRNA or IL-6; oligo, FCCP and AA were added at the time points indicated by dashed lines (n = 4). Error bars represent SEMs, * *p* < 0.05, *** *p* < 0.001, as determined by two-tailed Student’s *t*-test.

**Figure 7 ijms-23-10008-f007:**
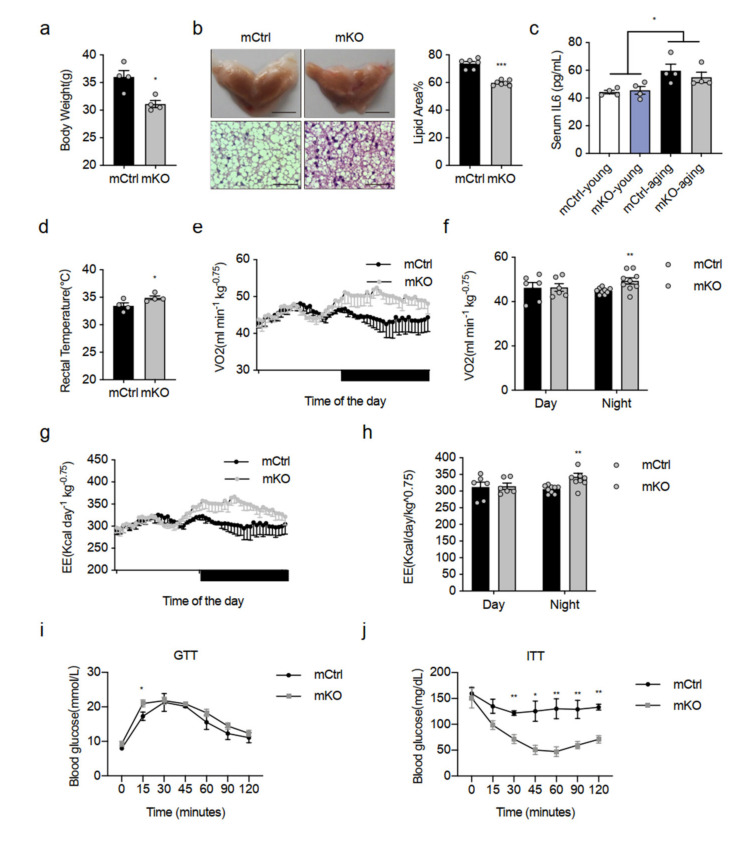
Deletion of Islr exerts beneficial metabolic effects in middle-aged mice: (**a**) body weights of the mCtrl and mKO mice at the age of 10 months (n = 4); (**b**) representative BAT tissues, H&E staining of BAT sections, and quantification of lipid area from mCtrl and mKO mice at the age of 10 months (n = 6–7); scale bar: 0.5 cm and 50 μm; (**c**) the level of serum IL-6 in young and middle-aged mCtrl and mKO mice (n = 4); (**d**) quantification of the core body temperatures of mCtrl and mKO mice (n = 4); (**e**,**f**) indirect calorimetric analysis of VO_2_ (**e**) and quantification of VO_2_ (**f**) of mCtrl and mKO mice at the age of 10 months (n = 6–9); (**g**,**h**) indirect calorimetric analysis of EE (**g**) and quantification of EE (**h**) of mCtrl and mKO mice at the age of 10 months (n = 6–9); (**i**) glucose tolerance test of the mCtrl and mKO mice at the age of 10 months (n = 5–6); (**j**) insulin tolerance test of the mCtrl and mKO mice at the age of 10 months (n = 4). Error bars represent SEMs, * *p* < 0.05, ** *p* < 0.01, and *** *p* < 0.001, as determined by two-tailed Student’s *t*-test.

## Data Availability

The data presented in this study are available in this article or Appendix A.

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
