# Peer review of "Immunoglobulin Superfamily Containing Leucine-Rich Repeat (Islr) Participates in IL-6-Mediated Crosstalk between Muscle and Brown Adipose Tissue to Regulate Energy Homeostasis"

_ijms, 2022, doi:10.3390/ijms231710008_

Round 1

Reviewer 1 Report

The manuscript of Liu et al. is aimed at investigating the role of immunoglobulin superfamily containing leucine-rich repeat (Islr) in the crosstalk events between brown adipose tissue and muscle and to reveal its relevance as a regulator IL-6-mediated signaling in metabolic homeostasis.  To achieve this goal, the Authors created several Islr-knock-out mice models and characterized their metabolic phenotype in physiological conditions and after cardiotoxin-induced injury. Authors employ an impressive variety of both in vivo and in vitro methods to reveal the biochemical and cellular signaling mediating the observed effects related to Islr deficiency. 

The topic fits the scope of the “International Journal of Molecular Sciences”, The manuscript present data in 7 multipanel figures. In addition, the manuscript is supplemented with 5 Supplementary Figures and 2 Supplementary Tables. In addition, the original images are provided for the blots presented in truncated forms in the manuscript.

The experiments are well designed and clearly described in the Methods section. The only part of the “Methods” section where more detailed description is needed is the part “4.1. Animals . Indeed, the hours of food restriction before the GTT and ITT experiments, the quantity of glucose and insulin administered, the glucometer type used for glucose measurements should be described. In addition, the type of food supplied to the mice (brand name + supplier) should be noted.  The Figures represent data that are scientifically sound.

The major issue with the manuscript is its presentation. Indeed, the manuscript is extremely difficult to read and to follow; it needs serious revision to improve clarity. All abbreviations should be spelt out on the occasion of first mentioning. The manuscript should be thoroughly revised to improve the logic of the flow of the text. Just as one example, no attempt was made to explain the mechanism of action and the organ targeted by cardiotoxin. Similarly, no rational is provided for the utilization of Myf5-Cre and Pax7-CreER mice to create tissue-specific Islr knock-out mice. 

The best advice this reviewer can give to Authors is that they should keep in mind that the “International Journal of Molecular Sciences” has a wide scope of interest and reaches scientists with different scientific background. Thus, some issues need to be explained in a more detailed fashion. 

Some selected points of specific concerns: 

1.     Title needs revision: please spell out “Islr” as “Immunoglobulin superfamily containing leucine-rich repeat (Islr)”. 

2.     Page 1 Line 16: spell out the abbreviation “Ndusf 2”

3.      Page 1, Line 26-27: “As a key site for nonshivering thermogenesis, brown adipose tissue (BAT) dissipates excess fuel energy as heat, and this dissipation counteracts obesity and its related metabolic disorders [3, 4].”

Please rephrase, it is an overstatement: BAT nonshivering thermogenesis does not “counteract obesity”, rather simply moderates the positive energy balance associated  with obesity.

4.     Page 1 Lines 34 and 35 : Spell out the abbreviation for  “IRF4” and “MSTN”. 

5.     Please check the manuscript for other non-described abbreviations and correct them, this is not a reviewer’s task to point out every such mistake. This remark also applies to the Figures (e.g. “5dpi” in Figure 1).  

 6.    Page 15, Lines 403: Authors state that mice were housed ambient temperature of 22+/- 2C. Several recent publications addressed the question of the effects of housing temperature on the translation value of results obtained in rodents to human conditions, most relevantly to BAT effects. E.g. :  Fischer et al.: The answer to the question “What is the best housing temperature to translate mouse experiments to humans?” is: thermoneutrality (Molecular Metabolism Vol. 26, August 2019, Pages 1-3). Authors should include a short paragraph in the “Discussion” part providing arguments against any conflicts in their results due to applied housing temperature. (And should reconsider housing temperatures for their further studies).

Reviewer 2 Report

The authors have conducted a comprehensive metabolic phenotyping of mice with deletion of Islr from whole-body and in BAT and skeletal muscle. They report that loss of Islr in BAT of mice subjected to skeletal muscle injury promotes enhanced thermogenic gene expression, increased energy expenditure, and enhanced mitochondrial function. The author’s conclusions are largely consistent with the data they present; however, some of the language should be edited for clarity and to be certain to not overstate the results presented here. I have divided my comments into major points that need to be addressed and minor points that could be edited but that do not detract from the work:

Major points:

Could the authors please provide more details for the cardiotoxin (CTX) experimental design? Please provide a catalog number or a more thorough description for the CTX that was used. Please cite any relevant references that also use this model. Please clarify the expected effects of CTX—is CTX known to increase IL-6? Is it known to lower thermogenesis in wild-type mice?

The authors should take care to eliminate the use of the phrase “BAT-specific” in reference to their genetic model. While I agree that most of the phenotype does seem to be driven by the BAT, it would be more accurate to change the language to save “loss of Islr in BAT” rather than “BAT-specific”. This also occurs in the Supplemental Figure 2 title. (also Lines 63, 297, 335)

The authors should eliminate the use of the phrase “lipid size.” Please indicate when talking about adipose depot size (gross morphology) or lipid droplet size (H&E staining and quantification) by using words like “adipose depot size” and “lipid droplet size”, respectively. (Examples: lines 52, 59, 79, 96, 119, 131, 212, 213, 306)

For the EE data presented in Figure 7, is there any data about activity during the dark phase? Are mKO mice more active and does that contribute to their increased EE? If so, this should be addressed.

Minor points:

Line 30: Please consider changing “the two most important” to “two of the most important”

Lines 34-36: Consider re-writing the summary of reference 10: Write about BAT-specific knockout of IRF4; instead of MSTN, please spell out myostatin.

Line 36: Instead of “a more recent study” perhaps “Another recent study” since these papers were published in the same year

Line 43: Instead of “proven” perhaps use the word “shown” or “demonstrated”

Line 51-52: Is there a citation for the group’s work on Islr KO in other contexts?

Line 59: “which led to enhanced EE”

Line 64: Instead of “advanced” perhaps “improved” or “enhanced”?

Line 72: get rid of the word “were”

Line 78: Instead of “lipid metabolism”, consider using the phrase “lipid accumulation”

Line 81: get rid of the word “having” (also in Line 120)

Line 88: “analyses” instead of “analyzes” (also in Line 200)

Line 94: Instead of “however” consider using the word “Importantly,” since this helps demonstrate that the changes in VO2 and EE are not attributable to changes in body weight or activity.

Line 124: Please add “in KO mice” to the end of this sentence to clarify that the observed changes are in the KO and not just due to CTX treatment.

Line 132: Mice is misspelled

Line 135: Figure 2 title: Consider using the phrase “increased energy expenditure” instead of “improved”

Line 180: Instead of “thermogenesis”, consider using the phrase “thermogenic gene expression” since this is the extent to which thermogenesis can be investigated in cultured cells

Line 191: Consider editing the title of Figure 4: Instead of “myokines”, perhaps this figure title should read “Secreted factors”? Consider also changing “thermogenesis” as before: “Secreted factors promote thermogenic program in Islr-deficient brown adipocytes”

Line 201: No hyphen needed in “…muscle derived from”

Line 205: Instead of “treated” perhaps “to treat”

Line 208: Instead of “invested” perhaps “investigated”

Line 219: Perhaps “KO BATSVF treated with IL-6 exhibited greater OCR as evidenced by…” instead of “was greater in the OCR”

Line 226: Consider editing the text to clarify your summary; perhaps something like this: “… IL-6 may act as a myokine that participates in CTX injury and increases systemic EE”

Line 238: BSA instead of BAS

Line 241: Section 2.6 title “Ndufs2 interacts with Islr…”

Line 242: Instead of “advanced” consider using the word “enhanced”

Line 301: Instead of “consequences” consider using the word “phenotypes”

Line 312-313: Please clarify the sentences about the GTT and ITT results in mKO mice. Instead of “insulin tolerance” in Line 313, consider using the phrase “insulin sensitivity”

Line 333: “In this study, we demonstrated that Islr, an important marker of mesenchymal stromal/stem cells, controls communication between BAT and skeletal muscle via IL-6”

Line 345: To this sentence, consider adding that the effects were also seen in aged mice: “…the effects on BAT activity were observed in the muscle injury model and in aged mice, which raised the question…”

Line 355-356: This citation appears to be a review rather than a “previous study”—please consider rewriting this sentence for clarity and to eliminate the “previous study” introduction to the sentence

Line 373: Instead of “mediates” consider using this sentence to summarize the direction of the effect that Islr has. For example, “Islr, as an important effector of IL-6 signaling, antagonizes IL-6-induced WAT or BAT browning, and loss of Islr enhances IL-6-activated thermogenesis”

Line 469-472: Is it possible to provide any more details about the secretome proteomics study in mouse muscle after CTX injury? How long after CTX was the muscle collected? Is it 5 days like the in vivo studies? Can any additional details about the proteomics platform be provided or a citation where these results may have been published?

Round 2

Reviewer 1 Report

The revised version of the manuscript of Liu et al. underwent corrections according to the some of the points raised by this reviewer.

Unfortunately, there are still some issues to be corrected, in majority concerning English language utilization and clarity. This reviewer finds that the text has still to be significantly revised before acceptation.

In the following points I mention some issues to draw attention to the most relevant problems.

-       English text revision is necessary, after revision some sentences now do not make any sense.

-       Spelling out of some abbreviations are still missing.

-       Authors “jump” in the text, mentioning new information without linking it to previously mentioned facts.

This reviewer provides some examples for these points; however, it is not his job to revise the paper instead of the Authors.

Authors should put more effort into this manuscript before it can be accepted.

Some examples for the issues mentioned above:

1. Page 1, Line 14: the corrected version now does not make any sense “  The BAT-specific ablation mice loss of Islr in BAT mice after cardiotoxin injury resulted in improved mi-15 tochondrial function, increased energy expenditure, and enhanced thermogenesis. “ 

Please revise again. 

2. Page 1, Lines 39-41: “Adult muscle responding to injury (like cardiotoxin (CTX)), SCs will be activated and fall into the rapidly proliferation, dif-40 ferentiation and fusion to repair damaged muscle progress [12, 13].” Please revise English language. Please add a sentence about the mechanism of injury action of cardiotoxin on muscle cells.

3. Please spell out : 12,13-diHOME

4. « Several studies have proven demonstrated that meteorin-like, a muscle-derived myokine, is induced after exercise, and this myokine stimulates beige fat thermogenesis-related gene expression, energy expenditure (EE) and improves glucose tolerance [21]. »  

This is a typical example where Authors suddenly “jump” to a new subject without any connecting sentence to previous sentences. Authors described BAT but had no mention about its relation to beige fat cells.

Round 3

Reviewer 1 Report

Authors addressed the issues raised by the reviewer in a satisfactory manner.